# Evaporation of Nanofluid Sessile Droplets Under Marangoni and Buoyancy Effects: Internal Convection and Instability

**DOI:** 10.3390/nano15040306

**Published:** 2025-02-17

**Authors:** Yuequn Tao, Zhiqiang Zhu

**Affiliations:** 1National Microgravity Laboratory, Institute of Mechanics, Chinese Academy of Sciences, Beijing 100190, China; taoyuequn@imech.ac.cn; 2School of Engineering Science, University of Chinese Academy of Sciences, Beijing 100049, China

**Keywords:** droplet evaporation, nanofluid, thermal convection, convection instability

## Abstract

Previous research has studied the evolution of patterns during the evaporation of sessile droplets of pure liquid, although there is a lack of reports focusing on the transition of flow regimes and flow stability of nanofluids. In this study, we investigate the evaporation of sessile droplets of Al_2_O_3_-ethanol nanofluid to elucidate the dynamic characteristics of the evaporation process from the perspective of internal convection. As the temperature increases, internal convection intensifies, significantly accelerating the evaporation rate. Three distinct convection flow patterns are observed under the combined influence of the Marangoni effect and buoyancy during evaporation: initially, two macroscopic convection cells form, followed by the periodic generation and propagation of hydrothermal waves (HTWs) near the contact line. Subsequently, Bénard–Marangoni (BM) convection cells gradually emerge and ultimately dominate the flow dynamics. The deposition patterns, which differ in part from the classic coffee-ring pattern, are closely related to the flow patterns of HTWs and BM convection cells during the pinning stage of droplet evaporation. Furthermore, the critical Marangoni (Ma) and Rayleigh (Ra) numbers for the onset of convection flow instability increase with rising substrate heating temperature.

## 1. Introduction

Droplet evaporation is not only common in nature and human life, but it also plays an important role in various thermal fluid devices, such as spray cooling equipment, heat pipes, and evaporators. Nanofluids, which are suspensions of nanoparticles within a base fluid, are recognized as promising candidates for high-performance heat transfer fluids. Investigating the evaporation characteristics of nanofluid droplets is essential for enhancing the application of nanofluids in thermal fluid processes. The internal convection and instability characteristics of nanofluid droplets exhibit notable differences compared to those of conventional fluids, which subsequently influence their evaporation rates and heat transfer efficiencies.

Many researchers have studied the evaporation of sessile droplets of nanofluids, most of which focus on the evaporation kinetics [1,2,3,4,5,6,7,8], wetting dynamics [1,3,7,9,10,11], and deposition patterns [2,3,8,12,13,14]. These findings indicate that droplets containing nanoparticles differ in evaporation dynamics and fluid motion from those without nanoparticles. However, whether the nanoparticles enhance or weaken evaporation and heat transfer is still unclear [15]. The complex convection pattern and the instability induced by the combined effects of Marangoni and buoyancy are contributing factors to the phenomenon mentioned above. Marangoni convection is associated with a gradient of surface tension. This gradient may be caused by temperature variations, leading to thermal Marangoni convection. Marangoni convection occurs when the variation in surface tension forces dominates over viscous forces. The Marangoni number, which is a dimensionless number, represents the ratio between the tangential stress and the viscosity, and can determine the strength of the convective motion [16]. Buoyancy convection occurs in the presence of gravitational acceleration due to density gradients induced by temperature differences within the liquid [17]. In an evaporating droplet, a tangential temperature gradient at the gas–liquid interface coexists with a normal temperature gradient, therefore inducing various types of convection, including Marangoni and buoyancy convection. As a result, the temperature gradients and characteristic length of droplets vary both spatially and temporally. Moreover, high evaporation rate would destabilize the convection inside the droplet and complicate the thermal patterns on the droplet surface [18].

The phenomenon of thermal convection instability within evaporating droplets of pure fluids has been investigated by many researchers in prior studies [19]. Using infrared thermography, Sefiane et al. [20,21] observed hydrothermal waves (HTWs) in evaporating droplets for the first time. HTWs are a type of unsteady and inherently three-dimensional traveling waves, composed of alternating cold and hot stripes in the direction of propagation. They also observed thermally driven convection cells in FC-72 droplets, and pointed out that the convection flow pattern may vary significantly for droplets of different liquids and on substrates of different conductivities. Carle et al. [22] used parabolic flight to explore the hydrothermal behavior of ethanol drops in microgravity condition. The findings demonstrate that the thermocapillary forces alone are responsible for the generation of HTWs. Wang and Shi [23] experimentally investigated the transition from the HTWs and stationary longitudinal rolls (SLRs) to the Bénard–Marangoni (BM) convection cells inside an evaporating droplet at constant contact line mode. HTWs and SLRs arise in moderately volatile droplets of methanol and highly volatile R-113 droplets, respectively, and are both replaced by BM convection cells when the contact angle decreases to the critical value. The tangential and normal Marangoni numbers were analyzed in detail to reveal the mechanism for the transition. It was also found that the substrate temperature greatly affected the Marangoni convection instability [24]. The Marangoni flow instability pattern of silicone oil drops with low volatility was investigated by Zhu and Shi et al. [18,25]. Unlike the traditional polygonal BM cells in a flat liquid layer, they pointed out that the cell in a droplet is naturally circular-arc, except for the straight connecting borders between cells. Zhong and Duan [26] studied HTWs in a steady-state evaporating droplet without shrinking. They continuously observed sustained hydrothermal waves, but the number of HTWs decreased over time due to evaporation. Zhang et al. [27] discovered, using numerical analysis, that a transition from axisymmetric to multi-cellular convection patterns occurred in a sessile water droplet due to the competition between Marangoni flow and evaporation.

The investigation of capillary convection induced by the evaporation of single-phase fluid droplets has established a foundational understanding for the examination of nanofluid droplets. In contrast to single-phase pure fluids, nanofluids exhibit distinct thermal properties and modes of energy transfer due to the influence of nanoparticles. Pallavi Katre [19] observed richer HTWs in nanofluids than in pure fluids. Using fractal dimension analysis, Wąsik et al. [17] found that the complex deposition patterns emerge from the dynamic process of the Bénard–Marangoni instability. Tao and Liu [18] found the evaporation of nanofluid droplets can be enhanced and the coffee-ring effect can be suppressed under buoyancy and Marangoni convection. The new forms of flow instability and non-equilibrium thermodynamic dissipation structures that may arise from thermal capillary convection in evaporating nanofluid droplets remain inadequately understood, and necessitate further investigation and exploration.

While previous research has examined the evolution of patterns during the evaporation of pure fluid sessile droplets, there is a scarcity of reports on the transition of flow regimes and stability in nanofluid droplets. This study looks into the patterns of convection instability and transition phenomena in a nanofluid sessile droplet during evaporation at the pinning stage. The evaporation process of sessile droplets on a heated substrate was experimentally studied through side view morphology observation and top view infrared observation. The deposition patterns of nanoparticles are examined using microscopic imaging techniques. The critical dimensionless numbers for unsteady flow patterns at various temperature conditions are determined. The experimental results of this article provide a basis for controlling the evaporation rate and enhancing the heat transfer efficiency of nanofluid droplets from the perspective of internal flow.

## 2. Experimental Setup

### 2.1. Experimental Setup

The schematic of the experimental setup is illustrated in Figure 1. The evaporation process of nanofluid droplets occurs on a horizontally positioned substrate, the design of which is depicted in Figure 2. To enable precise infrared observation, the surface was treated using a black anodizing process. A circular groove with an inner diameter of 4 mm is machined into the surface of the substrate, which extends the pinning stage of the droplets and establishes an initial contact line with a diameter of 4 mm. The selection of a contact radius larger than the capillary radius allows for a more effective study of buoyancy effects. Without this groove, droplets of the same volume would exhibit a larger diameter but a reduced height, resulting in a flattened shape that diminishes the buoyancy effect. Prior to each experiment, a droplet is deposited onto the substrate within the central pedestal using a micropipette (Eppendorf, 2~20 μL). The substrate is mounted on a flat plate heat exchanger. The substrate temperature is regulated by thermostatic water circulation from a constant temperature water bath (Manufacturer: HAAKE, Type: PHOENIX P1-C25P, Made in Karlsruhe, Germany) to the heat exchanger, with temperatures ranging from 5 °C to 80 °C. Two T-type thermocouples (Manufacturer: Omega, Type: 5TC-TT-T-36-36, Made in Michigan City, IN, USA) are employed to monitor the temperature of the substrate surface. A CCD camera (Manufacturer: Teledyne DALSA, Type: m128, Made in Waterloo, ON, Canada) captures the instantaneous morphology of the liquid droplets from a side view. Opposite the CCD camera is an LED background light source. An infrared camera (Manufacturer: InfraTec, Type ImageIR 8300, Made in Dresden, Germany) is utilized to record the evolution of temperature distribution on the surface of the droplet. After each experiment, the deposited nanoparticles are removed, and the substrate is cleaned through ultrasonication in a deionized water bath.

The flowchart of the experimental procedure and methodology is shown in Figure 3. As the experiment commenced, the substrate temperature was stabilized by circulating water through a flat plate heat exchanger. Subsequently, liquid was injected from the top to create an initial droplet diameter of 4 mm. During the evaporation process, the morphological changes of the droplets were recorded from the side view, while the temperature field variations were documented from the top view. Once the droplets had evaporated, the characteristics of the sedimentation pattern were observed. After cleaning the evaporation surface, the experiment was repeated.

### 2.2. Materials

Al_2_O_3_-ethanol nanofluid is purchased from the Chinese company SEEDIOR and the initial mass fraction is 1%. Before each experiment, the solution will be stirred using a magnetic stirrer to ensure the uniform dispersion of the solution. The solutions containing spherical nanoparticles of 10 nm in diameter are used in the experiment.

### 2.3. Analysis

Both CCD and infrared images are captured at a rate of five frames per second. The contact angle, volume, and height of the droplets are obtained by analyzing the side view CCD images. With infrared image analysis, the evolution features of convection flow patterns can be derived. A microscope (Manufacturer: Leica, Type: 2700 M, Made in Wetzlar, Germany) with a 5× lens is used to examine the deposition pattern. At least three samples are tested in each group to ensure the reliability of the results, and the typical cases are selected for analysis.

Two dimensionless numbers are introduced to qualitatively measure the effects of gravity and surface tension on the flow patterns transition:(1)$$Ma=-\frac{d\gamma }{dT}\cdot \frac{H\cdot \Delta T}{\mu \alpha }$$(2)$$Ra=\frac{v}{\alpha }\cdot \frac{g\beta H^{3}\Delta T}{v^{2}}$$
where γ is the surface tension coefficient, *H* is the height of the droplet, ∆*T* is the temperature difference of the heating substrate and the apex, *μ* is the dynamic viscosity of the fluid, *α* is the thermal diffusivity, *ρ* is the density, *g* is the acceleration of gravity, and *β* is the volume expansion coefficient. Note that the concentration and other physical properties of the nanofluid change as the evaporation goes on. These effects are taken into account when calculating the dimensionless numbers, as demonstrated in our earlier work [28].

## 3. Experimental Results and Discussion

Experimental investigations into convection instability patterns and transition phenomena in a sessile droplet of Al_2_O_3_-ethanol nanofluid during evaporation are conducted. Through side-view morphological analysis, we obtain the evaporation kinetics characteristics at various temperatures. The internal convection and instability characteristics of nanofluid droplets can significantly influence their evaporation rates. Different convection instability flow patterns and their transitions were observed using top-view infrared camera images. The corresponding deposition patterns are examined to further validate the characteristics of internal flow. The critical dimensionless numbers at different heating temperatures are subsequently analyzed.

### 3.1. The Evaporation Rate

Once the substrate reaches the desired temperature, we inject the liquid into the inner area of the groove to form a droplet. By processing each image, we obtained the changes in droplet volume, height, contact line radius, and contact angle over the evaporation period, under substrate temperatures ranging from 30 °C to 70 °C, as illustrated in Figure 4. The constant contact radius mode (CCR) begins at the initial stage of evaporation and occupies most of the evaporation time. Then the mixture evaporation mode follows. The volume and height of the droplets decrease nearly linearly over time. The impact of substrate temperature on the rate of evaporation is highly significant. The higher the temperature, the smaller the relative proportion of the constant contact radius phase.

### 3.2. The Convection Flow Pattern

The evolution of the surface temperature of the nanofluid droplet during evaporation is illustrated in Figure 5 with the substrate heating temperature of 45 °C. It can be seen that two flow convection cells first appear at the initial stage of droplet evaporation. Such cells were also observed in previous studies using CuO-water [3] and Al_2_O_3_-water [28] nanofluid. This convection pattern is mainly caused by the Marangoni-buoyancy effect. The non-uniform evaporation rate and heat conduction path of an evaporating droplet result in an inhomogeneous surface temperature distribution. The surface temperature close to the triple line region is higher than that of the apex, therefore Marangoni–Bénard convection forms inside the droplet as in Figure 6a. Additionally, heat transfer from the bottom to the top of the droplet causes a gradient in density, leading to a buoyant flow, depicted in Figure 6b. Consequently, thermal Marangoni–Bénard and Rayleigh–Bénard convection coexist and compete within the droplet. Figure 6 is intended solely as a conceptual representation and does not depict the actual streamline structure. The actual streamlines can be obtained using the Particle Image Velocimetry (PIV) method. There are also several numerical techniques available, including finite element methods, finite difference methods, and diffusion-based approaches. Initially, the buoyancy effect is significant at large droplet heights, but as the height decreases, the role of buoyancy weakens and the Marangoni effect becomes dominant. A source of flow instability arises around the contact line at 50.6 s. Then, from the source, a succession of waves travels along the contact line. The waves reach the furthest extent at 51.6 s and then begin to shrink in the propagation distance. At 52.9 s, the waves totally disappear, then other sources of instability start to develop, spread, and disappear. These sources are randomly generated near the triple-line region, where the evaporation rate is very high. The traveling-wave phenomenon occurs repeatedly during the evaporation process, with waves becoming increasingly wider. Previous researchers have also observed similar phenomena and believed that these waves are HTWs [23,29,30], which is a kind of Marangoni convection flow pattern caused by surface tension. However, the results of previous research on pure moderately volatile droplets show that HTWs are generated as soon as the droplet is deposited on the heating substrate and can quickly cover the entire contact line. These HTWs exist throughout the early stages of evaporation and do not disappear periodically. Therefore, it can be said that the addition of nanoparticles hinders the generation and propagation of HTWs. At 55.4 s, convection cells are generated, which are similar to BM convection cells [23,25]. Subsequently, more and larger convection cells are generated in the contact line region. The detachment of BM convection cells first occurs at 73.3 s. Afterward, these convection cells gradually develop toward the interior of the droplet. After 89.5 s, the depinning of the droplet contact line occurs, which is not the focus of this study.

The evolution of the diameter of BM convection cells with normalized time of *t/t_c_* under substrate temperature of 45 °C is shown in Figure 7, where *t_c_* is the lifetime of the nanofluid droplet with constant contact line diameter. The size of BM convection cells firstly increases and then decreases. At the beginning stage of evaporation, the convection cells inside the droplet continue to develop and become larger. During the later stages of evaporation, the thinning of droplet height causes a reduction in the size of convection cells.

Illustrated in Figure 8 is the flow pattern at the substrate temperature of 62 °C. The evolution process of flow patterns is similar to those under conditions of lower substrate temperature. Two large convective cells, HTWs, and BM convection cells appear sequentially during the evaporation process. Compared to lower heating temperatures, increasing the temperature extends the propagation distance of HTWs after they appear. HTWs spread until they cover the entire contact line. They do not disappear, but instead remain visible until BM convection cells arise. The BM convection cells, which can detach from the contact line, are generated at 23.8 s. At 27.2 s, the HTWs flow pattern is completely replaced by the pattern of BM convection cells. Compared to the cases at higher heating temperatures, at a reduced temperature of 30 °C, as shown in Figure 9, the instability of HTW flow does not occur in isolation; instead, it coexists with the flow pattern of HTWs and BM convection cells. Internal convection intensifies as the heating temperature increases, serving as a significant factor in the rapid escalation of the evaporation rate with rising temperature.

### 3.3. The Deposition Patterns

In an evaporating droplet, there are basically three kinds of internal flow: capillary compensation flow caused by rapid evaporation at the triple line, Marangoni flow caused by non-uniform surface tension, and buoyancy flow caused by density variation. The internal flow characteristics play a dominant role in the patterns of deposition [31]. The capillary flow in a pinned droplet leads to the formation of the well-known coffee-ring pattern. Marangoni flow towards the center of the droplet can mitigate the coffee-ring effect [28].

The typical deposition pattern after the drying of the Al_2_O_3_-ethanol droplet under different heating temperatures T*s* is listed in Figure 10. A qualitative analysis on the deposition patterns shows the reproducibility of the experimental results. As shown in Figure 8, the coffee-ring deposition pattern is inhibited in the current experiments, indicating that the Marangoni effect plays a dominant role. At a heating temperature of 30 °C, the flow pattern of BM cells appears during the whole duration of droplet evaporation; at higher heating temperatures of 45 °C and 62 °C, BM cells arise following HTWs. The ultimate deposition pattern in Figure 8 is the direct outcome of the progression of the aforesaid flow patterns. For different heating temperatures, the cell-like deposition pattern covers different areas corresponding to BM convection cells. The striped deposition pattern, which is formed in the region near the contact line, is directly related to the convection flow pattern of HTWs.

### 3.4. The Critical Values for Flow Instability

When the surface temperature of the droplet is observed to deviate from steady state and fluctuate in the top view infrared image, we believe the convection becomes unstable and dissipative structures are formed. The critical Rayleigh and Marangoni numbers are calculated when the flow field transitions from stable to unstable state. The volume, contact angle, and height are obtained by analyzing the side view image of the droplet. The thermophysical parameters of the nanofluid are obtained according to the previous research results. The critical Ma and Ra numbers for the inception of flow instability are plotted in Figure 11. As can be seen, both Ma and Ra numbers increase as the substrate heating temperature rises. Previous studies on pure fluids have shown a similar trend [32,33]. An increase in these two dimensionless numbers signifies greater convective intensity in the droplet, which accelerates its evaporation rate.

## 4. Conclusions

In the evaporation process of Al_2_O_3_-ethanol nanofluid sessile droplets with a pinning contact line of 4 mm in diameter, the convection flow pattern and features of flow instability are examined, and three different types of convection flow pattern are observed. The flow patterns of HTWs and the BM convection cells occur after the two-convection-cell flow pattern. Compared to pure fluid, the inclusion of nanoparticles is found to suppress the HTWs. The coffee-ring deposition pattern does not form under the combined effects of Marangoni and buoyancy. The pattern of deposition aligns with the way internal convection flow patterns evolve. The investigation of the influence of substrate heating temperature reveals that as the temperature rises, so do the critical dimensionless numbers for the commencement of flow instability. Enhancing droplet flow to improve evaporation is an effective strategy. Nanofluid evaporation is a very complicated process that involves constant changes in thermophysical characteristics together with multidirectional temperature gradients. Therefore, it is imperative to investigate the predominant mechanisms that give rise to different flow patterns in more detail. Further research is needed to fully understand the influencing parameters, which include the size, concentration, and variety of the nanoparticles.

## Figures and Tables

**Figure 1 nanomaterials-15-00306-f001:**
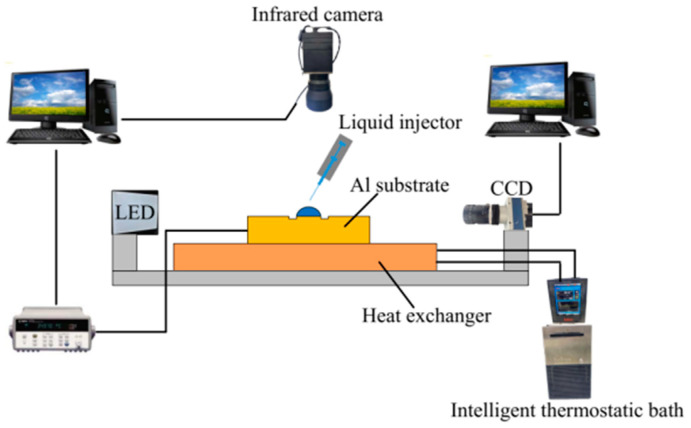
Scheme of the experimental setup.

**Figure 2 nanomaterials-15-00306-f002:**
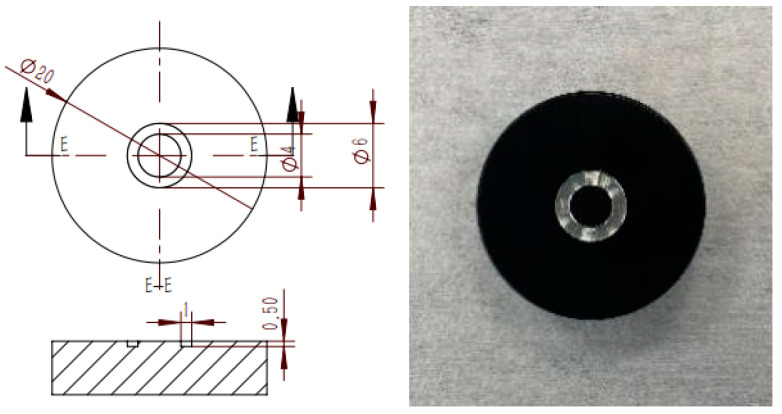
Detail design of the aluminum substrate.

**Figure 3 nanomaterials-15-00306-f003:**
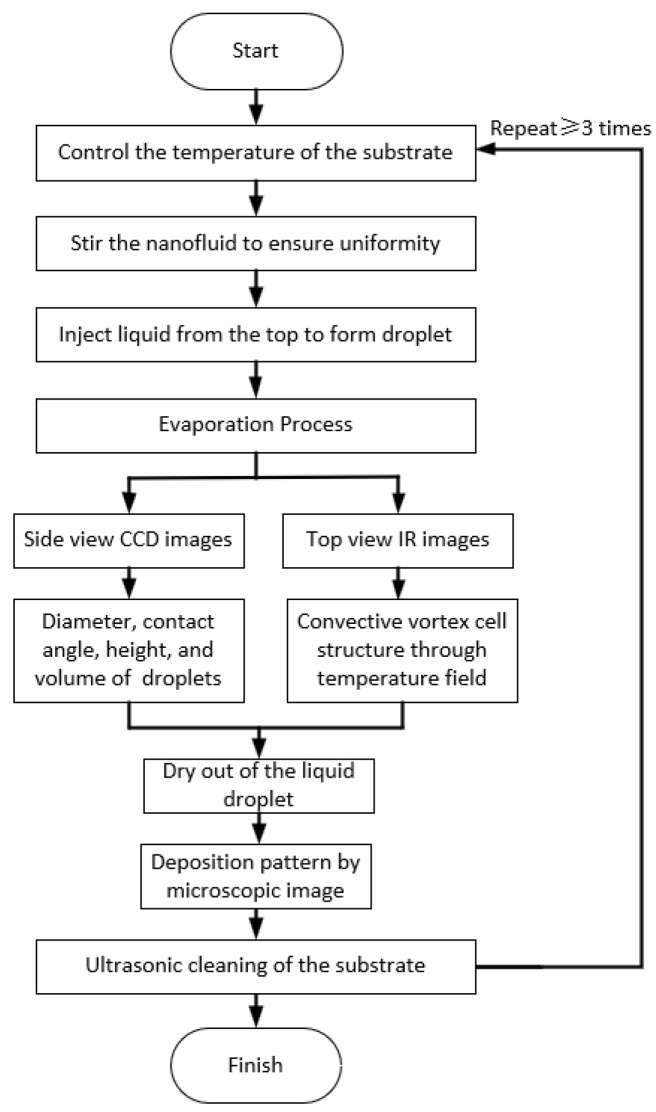
Flowchart of the experimental procedure and methodology.

**Figure 4 nanomaterials-15-00306-f004:**
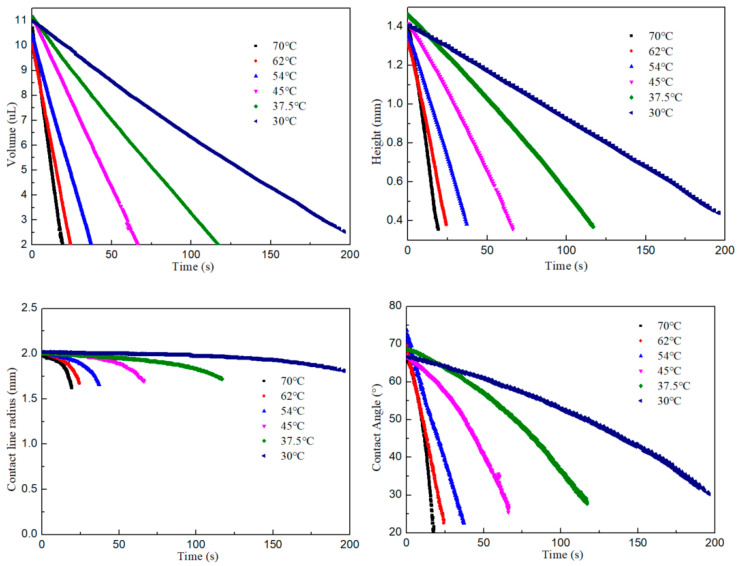
The variations of droplet parameters over evaporation time.

**Figure 5 nanomaterials-15-00306-f005:**
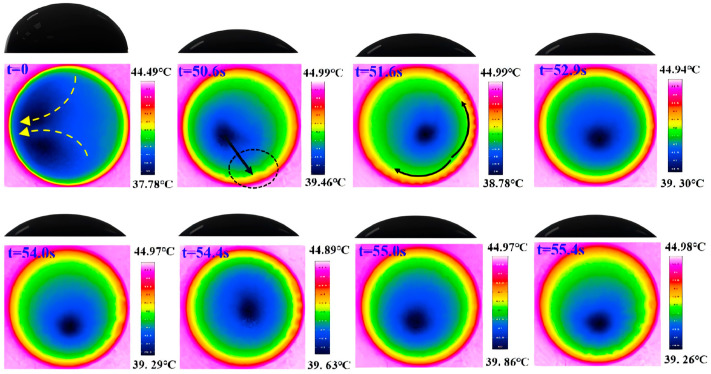
Top view infrared image and side view CCD image of a typical Al_2_O_3_-ethanol nanofluid droplet evaporation process (Ts = 45 °C, 10 nm, 1 wt%).

**Figure 6 nanomaterials-15-00306-f006:**
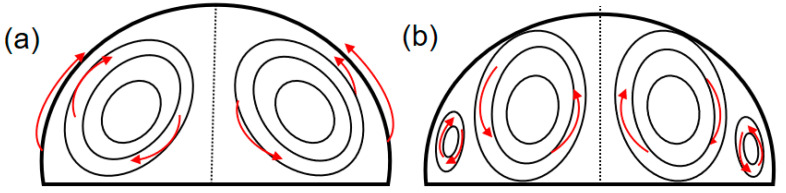
Schematic diagram of thermal Marangoni–Bénard and Rayleigh–Bénard convection. ((**a**). Marangoni-Bénard convection; (**b**). Rayleigh–Bénard convection; The red arrow indicates the direction of flow).

**Figure 7 nanomaterials-15-00306-f007:**
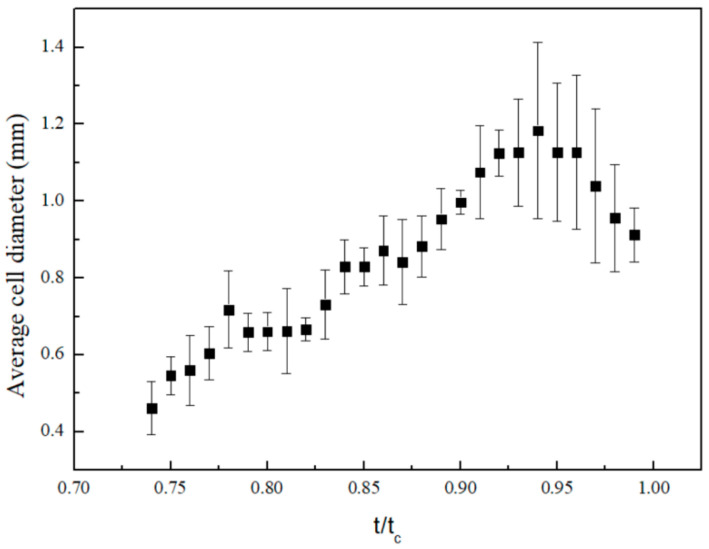
Evolution of the average diameter of BM convection cells with normalized time.

**Figure 8 nanomaterials-15-00306-f008:**
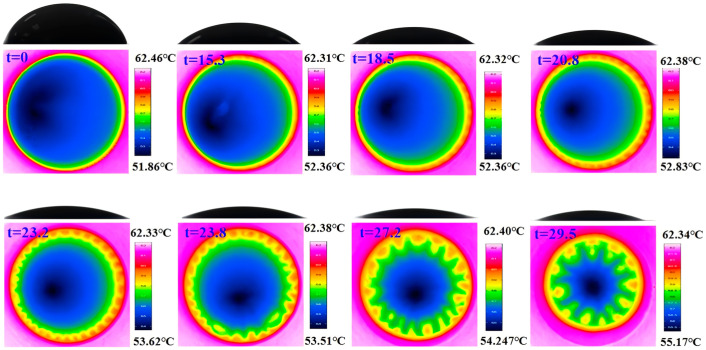
Top view infrared image and side view CCD image of a typical Al_2_O_3_-ethanol nanofluid droplet evaporation process (T_s_ = 62 °C, 10 nm, 1 w%).

**Figure 9 nanomaterials-15-00306-f009:**
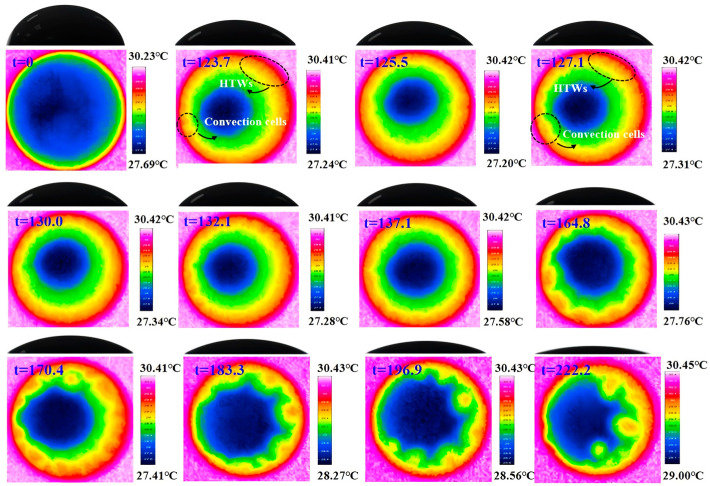
Top view infrared image and side view CCD image of a typical Al_2_O_3_-ethanol nanofluid droplet evaporation process (T_s_ = 30 °C).

**Figure 10 nanomaterials-15-00306-f010:**
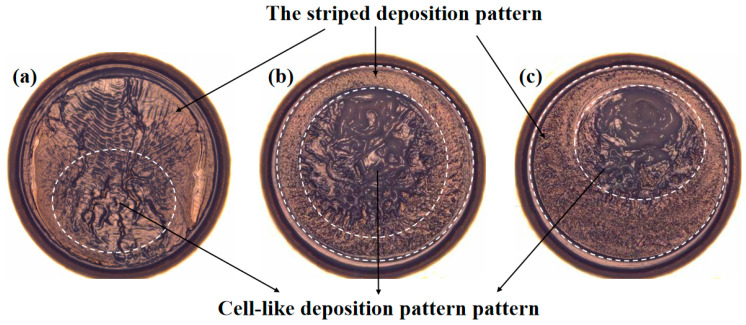
The deposition pattern: (**a**) T_s_ = 30 °C; (**b**) T_s_ = 45 °C; (**c**) T_s_ = 62 °C.

**Figure 11 nanomaterials-15-00306-f011:**
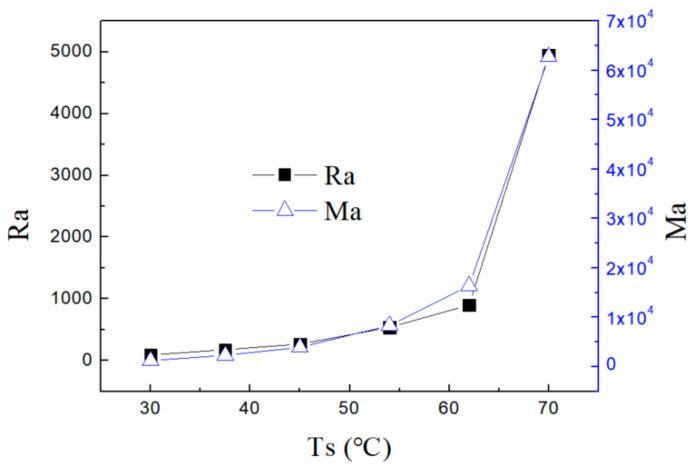
The critical dimensionless numbers for occurrence of thermal convection instability.

## Data Availability

Data will be made available on reasonable request.

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
