# Peer review of "Evaporation of Nanofluid Sessile Droplets Under Marangoni and Buoyancy Effects: Internal Convection and Instability"

_nanomaterials, 2025, doi:10.3390/nano15040306_

Round 1
Reviewer 1 Report
Comments and Suggestions for Authors
The paper experimentally describes the evaporation of a sessile nanofluid droplet with a particular focus on Marangoni and Buoyancy convection. The experimental apparatus is similar to the setup used by Tao and Liu (2023). Experiments determine the increasing surface temperature using an infrared camera and the geometry (decreasing height, volume, and radius of the droplet) using a CCD camera. The paper appears to be technically correct but lacks significant originality.
#1 In the introduction, basic regimes such as Marangoni and Buoyancy effects (mentioned also in the title, 10 times), HTWs (Hydrothermal Waves, 6 times), and BM (Bénard-Marangoni cells) are introduced but not explicitly defined. While it may not be strictly necessary, a brief definition of these four phenomena, including their physical interpretation and illustrative graphs (potentially with a constitutive equation for Marangoni flow), would be beneficial.
#2 The above remark also applies to Fig. 5, where contours of streamlines corresponding to Marangoni and Rayleigh cells are calculated. How were these contours derived? There are several numerical methods for calculating such streamlines from the experimentally obtained temperature profiles and cross-sectional geometries. The submitted paper mentions only fractal dimension analysis, but other possible methods include finite element methods, finite difference methods, or diffusional approaches. The same question applies to Fig. 6 regarding the calculation of the average size of BM convective cells.
#3 Most parameters in the study are constant: Al2O3 + water nanofluid at a fixed concentration, a 4 mm droplet diameter at the contact line, and the only variable parameter being the substrate temperature Ts (30, 45, 62 °C). I am concerned that using only one variable parameter might not suffice to define a general criterion for instability limits (e.g., Rayleigh and Marangoni dimensionless numbers). Was the critical Rayleigh number derived from velocity oscillations and the Marangoni number from temperature oscillations?
#4 What is the purpose and geometry of the circular aluminum substrate groove?
#5 Regarding Fig. 10, the last data point corresponding to Ts=70°C appears to be an outlier, as the highest measured substrate temperature is only 62°C (see Figs. 7 and 9).
Comments on the Quality of English Language
I am not a native speaker and therefore I cannot evaluate quality of english.
Author Response
Response to Reviewer
- Summary
Thank you very much for your valuable comments and suggestions, which have significantly improved the presentation of our manuscript. We have carefully considered all of your feedback. As you mentioned, we utilized equipment similar to that in our previous article (Tao and Liu, 2023). However, it is important to emphasize that in this article, we present significantly different research results, observing a more complex convective vortex cell structure and a transition process in flow patterns. Although there have been many studies on droplet evaporation, there is still a lack of reports focusing on the transition of flow regimes and flow stability of nanofluids. Based on the comments we received, the manuscript has been significantly revised by the authors. Careful modifications marked with red have been made to the revised manuscript. The point-by-point responses to your comments are as follows:
2. Point-by-point response to Comments and Suggestions for Authors |
Comments 1: In the introduction, basic regimes such as Marangoni and Buoyancy effects (mentioned also in the title, 10 times), HTWs (Hydrothermal Waves, 6 times), and BM (Bénard-Marangoni cells) are introduced but not explicitly defined. While it may not be strictly necessary, a brief definition of these four phenomena, including their physical interpretation and illustrative graphs (potentially with a constitutive equation for Marangoni flow), would be beneficial. |
Response 1: Thank you for your remark. We agree that clarifying these important definitions is crucial. In our original manuscript, physical interpretation and illustrative graphs of Marangoni-Bénard and Rayleigh-Bénard convection have been given (Page 6, lines 4~10; Page 7, Fig.5). In the introduction of the revised manuscript, we have added detailed physical interpretation of Marangoni convection and buoyancy convection(Page 2, lines 12~19). Comments 2: The above remark also applies to Fig. 5, where contours of streamlines corresponding to Marangoni and Rayleigh cells are calculated. How were these contours derived? There are several numerical methods for calculating such streamlines from the experimentally obtained temperature profiles and cross-sectional geometries. The submitted paper mentions only fractal dimension analysis, but other possible methods include finite element methods, finite difference methods, or diffusional approaches. The same question applies to Fig. 6 regarding the calculation of the average size of BM convective cells. Response 2: Thank you very much for bringing this to our attention. We would like to clarify that Figure 5 is intended solely as a conceptual representation and does not depict the actual streamline structure. Its purpose is to assist in explaining the convective vortex cell structure of the temperature field as observed from a top view. The actual streamlines can be obtained using the Particle Image Velocimetry (PIV) method. As the reviewer noted, there are also several numerical techniques available, including finite element methods, finite difference methods, and diffusion-based approaches. The points mentioned above have been incorporated into the revised manuscript(Page 6, lines 12~16). Comments 3: Most parameters in the study are constant: Al2O3 + water nanofluid at a fixed concentration, a 4 mm droplet diameter at the contact line, and the only variable parameter being the substrate temperature Ts (30, 45, 62 °C). I am concerned that using only one variable parameter might not suffice to define a general criterion for instability limits (e.g., Rayleigh and Marangoni dimensionless numbers). Was the critical Rayleigh number derived from velocity oscillations and the Marangoni number from temperature oscillations? Response 3: Thank you for your question. Both the two critical parameters are obtained through changes in the vortex cell structure of the top view temperature field. It is really true as Reviewer says that other factors, such as the concentration of nanoparticles and the diameter of the droplet contact line, can also influence the flow transition process of nanofluid droplets. Research involving additional parameters, numerical simulations, and quantitative observations, such as Particle Image Velocimetry (PIV) techniques, is necessary and planned to elucidate the mechanisms of flow instability. While previous research has studied the evolution of patterns during the evaporation of sessile droplets of pure liquid, there is a lack of reports focusing on the transition of flow regimes and flow stability of nanofluids, which play a very important role in the aforementioned application. We think whether researchers or engineers are able to find something valuable in the present manuscript. We wish that the reviewer can also agree on the value of the present manuscript.
Comments 4: What is the purpose and geometry of the circular aluminum substrate groove? Response 4: Thank you for your question. This groove is designed to create droplets with a consistent contact diameter each time. The selection of a contact radius larger than the capillary radius allows for a more effective study of buoyancy effects. Without this groove, droplets of the same volume would exhibit a larger diameter but a reduced height, resulting in a flattened shape that diminishes the buoyancy effect. The points mentioned above have been addressed in the revised manuscript.(Page 4, lines 2-5). Comments 5: Regarding Fig. 10, the last data point corresponding to Ts=70°C appears to be an outlier, as the highest measured substrate temperature is only 62°C (see Figs. 7 and 9). Response 5: Figures 7 and 9 do not correspond to the final point in Figure 10. As mentioned earlier (Page 5, lines 14–15), the temperature range of the experiment is 30–70°C. We present the results at 62°C in Figures 7 and 9. The flow pattern and transition process at 70°C are similar; however, the evaporation and transition rates are faster. |
Reviewer 2 Report
Comments and Suggestions for Authors
Evaporation of Nanofluid Sessile Droplets under Marangoni and Buoyancy Effects: The internal convection and instability
The idea of the manuscript is good but several changes must be implemented before it accepted. Right now, my suggestion is a major revision:
1. Abstract
1.1. This paragraph I strongly recommend to move to the introduction section: “The evaporation of nanofluid droplets has potential application value in enhancing heat transfer in thermal fluid systems. The internal convection and instability characteristics of nanofluid droplets exhibit significant differences when compared to those of conventional fluids, which in turn influence their evaporation rates and heat transfer efficiencies.”
1.2. The novelty of the manuscript must be added;
1.3. Numerical results must be added at the end of the abstract.
2. Introduction section
2.1. The contribution of the manuscript must be added at the end of this section;
2.2. Section 2 should be incorporated into Section 1;
3. Experimental apparatus section
3.1. A flowchart of the methodology applied must be added and explained with details;
3.2. The sensors and instruments specifications must be added;
3.3. There is no enough description and details about the experimental procedure conducted, so, this must be added with details;
4. Experimental Results and discussions section
4.1. A brief introduction must be added to explain what it will be showed and why;
4.2. Add the uncertainties from all the variables measure in the experimental tested;
4.3. Figure 3 needs more explanation and discussion.;
4.4. Figures 5 and 6, also need more explanation and discussion;
4.5. Equations 1 and 2 could be added in the experimental methodology, my suggestions;
4.6. Figure 10. The authors should be related the results from those numbers Ra and Ma), with the results presented in the literature and their own results, explain the similarities, limitations, and the challenges from the futures;
5. Conclusions section
5.1. This section must be improved regarding the indications requested.
Author Response
- Summary
Thank you very much for your valuable comments and suggestions, which have significantly enhanced the presentation of our manuscript. We have thoroughly considered all of your feedback. Careful modifications, highlighted in red, have been made to the revised manuscript. Below are our point-by-point responses to your comments:
2. Point-by-point response to Comments and Suggestions for Authors |
Comments 1: 1. Abstract:1.1. This paragraph I strongly recommend to move to the introduction section: “The evaporation of nanofluid droplets has potential application value in enhancing heat transfer in thermal fluid systems. The internal convection and instability characteristics of nanofluid droplets exhibit significant differences when compared to those of conventional fluids, which in turn influence their evaporation rates and heat transfer efficiencies.” 1.2. The novelty of the manuscript must be added; 1.3. Numerical results must be added at the end of the abstract. Response 1: 1.1. The paragraph is moved to the introduction section(Page 2, lines 3-5); 1.2. The novelty of the manuscript have be added (Page 1, lines 11-15) . 1.3. This article is an experimental research result, without numerical results. Comments 2: 2. Introduction section: 2.1. The contribution of the manuscript must be added at the end of this section; 2.2. Section 2 should be incorporated into Section 1; Response 2: 2.1. The contribution of the manuscript must be added at the end of this section(page 3, lines 38-40). 2.2. The content of the second section is briefly summarized in the first section(Page 3, lines 34-37). Comments 3: 3. Experimental apparatus section: 3.1. A flowchart of the methodology applied must be added and explained with details; 3.2. The sensors and instruments specifications must be added; 3.3. There is no enough description and details about the experimental procedure conducted, so, this must be added with details; Response 3: 3.1. and 3.3. A flowchart of the experimental procedure and analysis methodology applied and the description are added in the revised manuscript(Page 5, Fig.3, lines 3-10). 3.2. The sensors and instruments specifications have been added(Page 4, lines 14,17,18,20,22). Comments 4: 4. Experimental Results and discussions section: 4.1. A brief introduction must be added to explain what it will be showed and why; 4.2. Add the uncertainties from all the variables measure in the experimental tested; 4.3. Figure 3 needs more explanation and discussion. 4.4. Figures 5 and 6, also need more explanation and discussion; 4.5. Equations 1 and 2 could be added in the experimental methodology, my suggestions; 4.6. Figure 10. The authors should be related the results from those numbers Ra and Ma), with the results presented in the literature and their own results, explain the similarities, limitations, and the challenges from the futures; Response 4: 4.1. A brief introduction are added at the beginning to explain what it will be showed and why.(Page 6, lines 24-32). 4.2. We are sorry that some infrared and microscopic images in the article cannot add error bars. 4.3. We have added further explanations and discussions(Page 7, lines2-6) in the revised manuscript regarding Fig. 3 (now changed to Fig. 4). 4.4. We have added further explanations and discussions(Page 8, lines 15-24; Page 9, lines 10-14) in the revised manuscript regarding Fig. 5 and Fig.6 (now changed to Fig. 6 and Fig.7). 4.5. Equations 1 and 2 have been added in the section 2.3(Page 6, lines 11-22). 4.6. This result is qualitatively consistent with the trend in the literature(Page 13, lines 6-7). Comments 5: Conclusions section: 5.1. This section must be improved regarding the indications requested. Response 5: We are committed to continuously improving in accordance with the feedback provided by reviewers and editors. |
Round 2
Reviewer 2 Report
Comments and Suggestions for Authors
The paper is ready now